EU protected area network did not prevent a country wide population decline in a threatened grassland bird

Silva João P. jpsilva@cibio.up.pt silvaj@sapo.pt 1 2 3
Correia Ricardo 4 5 6
Alonso Hany 7
Martins Ricardo C. 1 2
D’Amico Marcello 1 2
Delgado Ana 2
Sampaio Hugo 8
Godinho Carlos 9
Moreira Francisco 1 2
1 REN Biodiversity Chair, CIBIO/InBIO—Research Centre in Biodiversity and Genetic Resources, University of Porto , Vairão , Portugal
2 CEABN/InBIO—Centre for Applied Ecology “Prof Baeta Neves”, Institute of Agronomy, University of Lisbon , Lisbon , Portugal
3 cE3c—Centre for Ecology, Evolution and Environmental Changes, Faculty of Sciences, University of Lisbon , Lisbon , Portugal
4 Institute of Biological and Health Sciences, Federal University of Alagoas , Maceió , Alagoas , Brazil
5 School of Geography and the Environment, University of Oxford , Oxford , United Kingdom
6 CESAM—Centre for Environmental and Marine Studies, Department of Biology, University of Aveiro , Aveiro , Portugal
7 CIBIO/InBIO—Research Center in Biodiversity and Genetic Resources, Pólo de Évora, University of Évora , Évora , Portugal
8 SPEA—Sociedade Portuguesa para o Estudo das Aves , Lisbon , Portugal
9 ICAAM—Instituto de Ciências Agrárias e Ambientais Mediterrânicas, Universidade de Évora, Laboratório de Ornitologia , Évora , Portugal
Gandini Patricia
Electronic publication date: 2018 Jan 23
Publication date: 2018
Volume: 6
Electronic Location ID: e4284
Received 2017 Oct 13; Accepted 2018 Jan 1
Copyright: ©2018 Silva et al.
Copyright year: 2018
Copyright holder: Silva et al.
License: This is an open access article distributed under the terms of the Creative Commons Attribution License, which permits unrestricted use, distribution, reproduction and adaptation in any medium and for any purpose provided that it is properly attributed. For attribution, the original author(s), title, publication source (PeerJ) and either DOI or URL of the article must be cited.
License URL: https://creativecommons.org/licenses/by/4.0/

Keywords: Farmland birds, Effectiveness, Little bustard, Natura 2000, Protected area network

Funding: European Union LIFE project LIFE02NAT/P/8476 Fundação para a Ciência e Tecnologia (FCT) SFRH/BPD/111084/2015 IF/01053/2015 The first national little bustard survey was funded by an European Union LIFE project (LIFE02NAT/P/8476) and carried out by the Portuguese Nature Conservation Institute (ICNF—Instituto da Conservação da Natureza e da Biodiversidade). The 2016 survey was coordinated by the REN Biodiversity Chair/CIBIO with the collaboration of ICNF, Liga para a Protecção da Natureza, Quercus—Associação Nacional de Conservação da Natureza, Sociedade Portuguesa para o Estudo das Aves, LABOR—Laboratório de Ornitologia/University of Évora. João P. Silva and Francisco Moreira were funded by grants SFRH/BPD/111084/2015 and IF/01053/2015 from Fundação para a Ciência e Tecnologia (FCT). The funders had no role in the analysis, decision to publish, or preparation of the manuscript.

==============================
Background

Few studies have assessed the effectiveness of the Protected Area networks on the conservation status of target species. Here, we assess the effectiveness of the Portuguese Natura 2000 (the European Union network of protected areas) in maintaining a species included in the Annex I of the Bird Directive, namely the population of a priority farmland bird, the little bustard Tetrax tetrax.

Methods

We measured the effectiveness of the Natura 2000 by comparing population trends across time (2003–2006 and 2016) in 51 areas, 21 of which within 12 Special Protection Areas (SPA) that were mostly designated for farmland bird conservation and another 30 areas without EU protection.

Results

Overall, the national population is estimated to have declined 49% over the last 10–14 years. This loss was found to be proportionally larger outside SPA (64% decline) compared to losses within SPA (25% decline). However, the absolute male density decline was significantly larger within SPA .

Discussion

In spite of holding higher population densities and having prevented habitat loss, we conclude that Natura 2000 was not effective in buffering against the overall bustard population decline. Results show that the mere designation of SPA in farmland is not enough to secure species populations and has to be combined with agricultural policies and investment to maintain not only habitat availability but also habitat quality.

Introduction

Protected areas constitute key tools for conserving biodiversity (Marton-Lefèvre, 2014; Watson et al., 2014). The European Union (EU) has set up the largest coordinated network of protected areas in the world—Natura 2000. Covering 18% of EU’s land area, it comprises Special Protection Areas (SPA) and Special Areas of Conservation (SAC) designated respectively under the Birds Directive (2009/147/EC) and the Habitats Directive (1992/43/EEC). EU countries are required to manage Natura 2000 sites to maintain or improve the conservation status of species and habitats listed in these Directives. Therefore, monitoring species populations, particularly those from target species, is essential to evaluate the effectiveness of Natura 2000.

Most research evaluating the effectiveness of the Natura 2000 protected area network has focused on assessing its spatial coverage of biodiversity values (e.g., Cabeza, 2013; Abellán & Sánchez-Fernández, 2015), but much less is known about its effect on species persistence or population trends (e.g., as a consequence of habitat loss or climate change) due to a lack of temporal data (e.g., Pellissier et al., 2013). Most recent studies compare population trends across species with different conservation priority levels (e.g., Sanderson et al., 2016), while rigorous studies based on empirical designs to compare conservation outcomes in areas with and without exposure to conservation policy instruments are still scarce (Miteva, Pattanayak & Ferraro, 2012). Even for taxa with abundant information on population sizes, such as birds, few studies have assessed the effectiveness of the SPA network on the conservation status of target species (Orlikowska et al., 2016).

Farmland birds are of high conservation concern in Europe, showing a steep population decline across the continent mainly due to ongoing intensification of agricultural practices (BirdLife International, 2004; Donald et al., 2006). At the European scale, however, few studies assessed whether SPAs designated in agricultural land are delivering any positive effects on farmland bird populations (Gamero et al., 2017). For example, Pellissier et al. (2013) used a national breeding bird survey to contrast species population trends in the period 2001–2010 in plots located within and outside Natura 2000 sites in France and found no significant differences. They concluded that the network of protected areas was established too recently to allow an assessment of its influence on population trends. In contrast, and at a larger scale, Gamero et al. (2017) used country wide information for a set of 25 EU countries. This study reported that species listed in Annex I of the Birds Directive (species with higher conservation status and for which EU Member States are obliged to implement special conservation measures) had higher population growth rates (during 1981–2012) in countries with a higher proportion of land designated as SPA. They conclude that EU policies seem to generally attenuate the declines of farmland bird populations, but do not reverse them.

The Portuguese farmland bird SPA network was created between 1994 and 2008, and consists of 13 areas covering over 195,000 ha which were delineated based on detailed information on the spatial patterns of occurrence of priority species listed in Annex I. Here, we assess the effectiveness of this SPA network in maintaining the population of a priority farmland bird, the little bustard Tetrax tetrax, a species that has undergone a major decline and breeding range reduction since the beginning of the last century (Iñigo & Barov, 2010). We expected that populations within SPA would show more favourable trends when compared to populations in farmland areas outside SPAs. For this purpose, we compared estimates of breeding population size and density over a decade (2003–2006 to 2016) within and outside SPAs. We further assessed trends in the availability of grasslands comprising fallow lands and extensive pastures, the preferred breeding habitat of the species (Morales, García & Arroyo, 2005; Silva, Palmeirim & Moreira, 2010; Moreira et al., 2012).

Methods

Study areas

The large majority of the little bustard population in Portugal is concentrated in Alentejo, Southern Portugal (Equipa Atlas, 2008). A first population survey was carried out in the region during 2003–2006 in the scope of a EU LIFE project (LIFE02NAT/P/8476) (Silva et al., 2006). This first survey was based on a network of 81 survey areas, from which 21 were located within 12 SPAs and the remaining 60 were outside SPAs. Each SPA was considered as a single survey area, with the exception of the three largest areas (over 10,000 ha of farmland) which accounted for between two and six survey areas. The size of SPA survey areas ranged from 1,715 and 4,718 ha (mean = 3,025 ha). Non-SPA survey areas consisted of approximately 2,500 ha quadrats defined as follows: (i) on a first stage, we delimited 10 × 10 km UTM quadrats whose land surface was covered by more than 40% of open agricultural and pastoral land area (representing the potential habitat for the species) based on information from Corine Land Cover 2000. These quadrats overlapped to a great extent with the quadrats where the presence of the species was recorded in the Portuguese Breeding Bird Atlas (Equipa Atlas, 2008); (ii) on a second stage, a maximum of two 5 × 5 km areas within each of the 10 × 10 km UTM quadrats identified in the first stage were randomly selected for surveying. Ten additional areas with potential for the species but not fitting these criteria were also surveyed (see Supplemental Information 1). Overall, a total of 60 non-SPA areas, stratified across the four main sub-regions of Alentejo (Fig. 1) were censused during this first period (2003–2006).

Figure 1 Location of the study area within Europe and Portugal.

(A) Location of Alentejo, the study area, within Europe. The polygons outlined in black show the network of key conservation areas for farmland birds, classified as SPAs. Dark grey areas indicate little bustard survey areas within SPAs. The white areas represent the survey areas outside SPAs but within potential habitat for the little bustard, which were stratified across the four sub-regions of Alentejo; Alto, Centro, Baixo and Litoral. (B) Example of a network of survey points placed along available dirt tracks within a survey area and used to estimate the density of breeding males.

In 2016, all previously surveyed areas within SPAs were revisited but, in non-SPA areas, the number of surveyed areas in each 10 × 10 km quadrat was reduced to just one 5 × 5 km area due to logistic constraints. The final result was a total of 51 survey areas sampled in both periods (Fig. 1), of which 21 were located within SPAs designated for farmland bird conservation and 30 were non-SPAs areas with potential habitat for the species.

Little bustard counts

Little bustard population densities were censused using a standardized protocol based on estimating male densities. In each of the survey areas we estimated male density from a network of survey points previously defined along dirt tracks and distanced by 600 m from each other and from disturbance factors, such as paved roads or inhabited houses (more details in Moreira et al., 2012). The dirt tracks were covered by car in early morning and late afternoon during April–May and each point was surveyed during 5 min within a prospecting radius of 250 m to detect little bustard males. Because of ample habitat availability, the location of survey points in SPAs fall mostly in farmland habitats, but in non-SPAs, some survey points could be located in unsuitable habitats (e.g., forest, scrubland). At each survey point the proportion of fallow land and pastures, hereafter designated grasslands, was assessed by estimating visually, within eight equal sections of the 250 m buffer, the number of sections where this land use was dominant. This method allowed for a rough estimation of the available main breeding habitat for the species (Morales, García & Arroyo, 2005; Silva, Palmeirim & Moreira, 2010; Moreira et al., 2012).

Almost all points surveyed during the 2003–2006 period were surveyed again in 2016. The points that became inaccessible in 2016 (<1% of the sampled points) were replaced by new ones, following the same requisites as mentioned above. The network of survey points covered each survey area at an average density of 0.96 points/km2 (median = 1, range = 0.33–2.12). Overall, a total of 2,326 and 1,441 survey points were sampled in 2003–2006 and resampled in 2016, respectively.

Data processing and analysis

Population estimates

For each survey area and sampling period, mean male density (and 95% confidence intervals) was estimated from the number of males found within the 250 m buffer of sampled points. The population estimate for each site was then calculated by extrapolating the mean density calculated from the survey points to the total area of potential habitat when within an SPA or to the whole survey area if outside SPA (see Supplemental Information 1 for a detailed description). For large SPAs with more than one sampled area, the mean density was obtained by calculating the average density across areas. We assessed the proportional increase or decrease in the estimated population sizes for the two sampled periods, across the two types of areas (SPA or non-SPA).

Density and grassland habitat differences between surveys and protection status

For the subset of 51 areas sampled in both periods, we modelled the effect of protection status (SPA versus non-SPA) and survey period (2003–2006 or 2016) on population densities (males/km2) and on the amount of suitable grassland habitat (km2). The amount of available grassland habitat was calculated multiplying the surface of each survey area by the estimated proportion of this land use derived from field estimates. We used generalized linear mixed models (GLMMs) with Gaussian distribution and identity-link function, implemented in package lme4, for both little bustard male density and grassland habitat availability. Protection status and survey period (including interactions) were considered as fixed effects in the model and survey area was included as a random effect to account for lack of independence within areas. Model fit was assessed using conditional and marginal r2 values (Nakagawa & Schielzeth, 2013). All analyses were implemented in R Software v3.1.3 (R Core Team, 2017) and figures were elaborated using the ggplot2 library available for the same software package.

Results

The results show an overall decline of breeding little bustards across most of its distribution range. The Portuguese national population estimate in 2003–2006 was of 17,418 males (95% CI [13,074–21,762]; Table S1). In 2016, it was estimated at 8,900 males (95% CI [5,008–12,836]; Table S1), representing an overall national decline of 48.9%. The amount of losses was greater outside SPA (10,724 to 3,892 males; −63.7%) compared to SPA areas (6,695 to 5,008 males; −25.2%).

In the subset of areas monitored in both surveys, results show a higher male density in SPA survey areas compared to non-SPA survey areas and a strong decline in density between surveys (Fig. 2). GLMM results showed a significant interaction between survey period and protection status (Table 1) indicating that, in absolute terms, density declines were even larger within SPA. Patterns for grassland availability showed higher habitat availability inside SPA and a higher habitat loss across time outside SPA (Fig. 2; Table 1).

Figure 2 Effects of protection status (SPA versus non-SPA) and survey period (2003–2006 or 2016) on population densities and on the amount of suitable grassland habitat.

Comparison of: (A) male little bustard densities (males/km2; mean and ± standard error) and (B) grassland area (males/km2; mean and ± standard error), within SPAs and outside SPAs (non-SPAs) in 2003–2006 and 2016.

Table 1 Summary statistics of the models exploring little bustard density and grassland availability between surveys and area protection status.

The reference level represents SPAs in the 2003–2006 survey. Conditional and marginal r2 values were 0.65 and 0.36 for the little bustard density model and 0.68 and 0.09 for the grassland availability model.

Variable	Estimate	Std. Error	p-value	
Little bustard density model	
Intercept	4.571	0.414	<0.001	
Non SPAs	−1.938	0.412	<0.001	
2016 Survey	−3.215	0.557	<0.001	
Non SPAs : 2016 Survey	1.191	0.581	0.045	
Grassland availability model	
Intercept	14.143	1.997	<0.001	
Non SPAs	−2.601	2.604	0.321	
2016 Survey	1.497	1.677	0.135	
Non SPAs : 2016 Survey	−4.895	2.187	0.030	

Discussion

A generalised population collapse

The 2016 survey showed a drastic 49% decline compared to 2003–2006. In 2003–2006 the conservation status of the species was mostly favourable across its range, and exceptionally high breeding densities were found in many SPAs, among the highest recorded for the species (Silva et al., 2014). The overall average density outside SPAs was also relatively high when compared with the densities reported across its western range (De Juana & Martínez, 1996; Jiguet, Arroyo & Bretagnolle, 2000).

Significant declines of little bustard populations have, in the past, been linked to land use change and habitat loss as consequence of agricultural intensification. That was the case of the French population that in the late 20th century, over a 17-year period, experienced a very rapid and dramatic decline of over 90% (Inchausti & Bretagnolle, 2005). More recently, declines have been reported for Spain, with over 50% loss of regional populations within the last decade (De Juana, 2009; Mañosa et al., 2015; Morales, Traba & Arroyo, 2015). Our results suggest that a similar population decline occurred in Portugal over the last few years.

The fact that the Portuguese population decline within SPAs was proportionally smaller when comparing the with the remaining areas, suggests that this network of protected areas managed to somewhat buffer the strong overall decline of the species. However, in absolute terms, the decline in male densities was stronger within SPAs, showing that the current management is not being able to maintain historically high densities. Despite this decline, SPAs still hold important populations, showing a higher density compared to areas outside SPAs (Fig. 2A). In addition, SPA were more effective in maintaining habitat availability than unprotected areas (Fig. 2B).

Even though this work is based on two single national surveys that took place 10 years apart, some areas were surveyed more times. Such was the case of a national SPA survey in 2010. Castro Verde, the most important breeding area for the species, that has also been counted every year since 2002 (Delgado & Moreira, 2010; A Delgado, pers. comm., 2017). Altogether these intermediate results indicate a population trend that is in agreement with the observed differences in the two-time periods.

Why did Protected Areas fail to prevent population decline of a target species?

The network of Natura 2000 SPAs designated for grassland bird conservation covers all Portuguese regions holding the most important little bustard populations (Silva et al., 2006). This is also confirmed by the fact that population densities in 2003–2006 were much higher in SPAs than in non-SPAs. However, coverage of protected areas by itself may be insufficient to ensure a favourable conservation status of target species (Watson et al., 2014), and effective management must be put in place to ensure the biological needs of these species (Leverington et al., 2010).

Taking into account the factors driving negative little bustard population trends in other regions of Europe (Iñigo & Barov, 2010), habitat loss would be expected to be the main driver of population decline, both outside and within SPAs. However, our indicator of grassland habitat availability, although somewhat crude, suggests that SPAs did manage to maintain the amount of habitat over time, whereas habitat loss continued to occur outside SPAs (e.g., with the expansion of irrigated and permanent crops). Natura 2000 areas benefit from legal mechanisms that prevent structural land use changes, for example interdicting the conversion of farmland to forestry, or preventing the installation of infrastructures (e.g., roads, transmission power lines, buildings) without impact assessment and compensatory measures. In spite of this legal protection and its effectiveness in maintaining habitat availability, significant breeding density losses were recorded in these areas.

A probable explanation to population declines within SPAs is that they did not manage to maintain habitat quality. Permanent pastures, which increased by up to 41% over a 10-year period as result of EU Common Agricultural Policies (CAP) incentives (INE, 2011; Ribeiro et al., 2014), have been progressively installed in previously extensively managed grasslands across the species range. Livestock density has also increased significantly over a similar period (Pimenta, Fernandes & Minhoto, 2015). This intensification process may potentially affect vegetation structure and ultimately grassland quality for the little bustard, particularly for nesting females that require taller vegetation (Morales et al., 2008; Silva et al., 2014), or still expose individuals or families to predation, due to the less dense sward structure and shorter vegetation (Tarjuelo et al., 2013; Silva et al., 2014). It may also impact food availability which is crucial for the development of chicks particularly during the first weeks of life (Jiguet, 2002). Although these management changes occurred both within and outside SPA, their impact is expected to be stronger in higher quality areas. The importance of setting an adequate grassland management scheme is highlighted by the fact that two important SPAs for the species, that maintained or increased their populations (Castro Verde and Vale do Guadiana; Table S2), were the ones where a significant number of farmers joined an existing agri-environmental program promoting extensive agricultural practices and establishing thresholds for grazing intensity (Pinto, Rocha & Moreira, 2005; Santana et al., 2014).

Another explanatory hypothesis is that generalized habitat degradation outside SPAs, nearby key conservation areas, may have led to larger post-breeding migration movements (Silva, Faria & Catry, 2007; García de la Morena et al., 2015), exposing bustards to higher levels of energy expenditure and to a greater mortality risk with anthropogenic infrastructures (Silva, Faria & Catry, 2007). Climate warming may also be impacting grassland quality by drying prematurely the vegetation and consequently limiting trophic resources. High anthropogenic mortality has been recently found in Iberia, with mortality annual rates estimated between 3.4 and 3.8% due to collisions with power lines and another 2.4–3% due to illegal killing, possibly contributing to the depletion of the population (Marcelino et al., 2017). Even though other factors are likely to be contributing to the decline of the little bustard, it is within the few SPAs with successful management schemes that breeding densities are stable or incrementing. The fact that some populations are incrementing and other declining suggests that birds can shift between breeding areas, performing movements towards better conserved habitat. Causal factors influencing site-level variability in population trends will be subjected to a further in-depth analysis including changes in habitat availability, land use cover and the prevalence of anthropogenic infrastructures.

Conclusion

We conclude that despite the importance of SPAs still holding important breeding populations and grassland habitat, the network of protected areas was not effective in buffering against the bustard population decline. This case study shows that the mere designation of SPAs in European farmland may not be enough to secure species populations and has to be combined with management (mainly agricultural) policies and investment directed at maintaining not only habitat availability but also habitat quality.

Supplemental Information

Supplemental Information 1 Additional description of the survey areas and population estimates

Click here for additional data file.

Table S1 Results of the two national surveys

Results of the two national surveys, presenting the mean, minimum (min) and maximum (max) total male estimates for SPA and non-SPA, also indicating the proportion of variation and difference of number of males.

Click here for additional data file.

Table S2 Results from the 2003–2006 and 2016 SPA surveys

Results from the 2003–2006 and 2016 SPA surveys, indicating the mean, minimum (min) and maximum (max) male density and male population estimates. The trend (considering stable populations with variations up to 10%), proportion of population variation and difference in number of males between surveys for each SPA are also presented.

Click here for additional data file.

Data S1 Raw data

Click here for additional data file.

We are indebted to all that carried out the field work: Nuno Sequeira, José Paulo Martins, Fernando Abegão, Joaquim Pífano, Joana Alves, Domingos Leitão, Carlos Cruz, Ivan Kljun, Rui Morgado, Pedro Salgueiro, Luís Venâncio, Henrique Velez, Rui Pedroso, Nuno Faria, Rita Alcazar, Carlos Pacheco, Bruno Martins, Ricardo Silva, Paulo Marques, Hugo Lousa, Pedro Rocha, Fernando Queirós, João Carlos Claro, Carlos Carrapato, Cristina Cardoso, Teresa Silva, Fernanda Romba, Eunice Pereira, Ana Martins, Célia Medeiros, Carlos Franco, Conceição Conde, Pedro Alverca, Agostinho Tomás, Raquel Ventura, David Carvalho, Pedro Capa. We would also like to acknowledge the review of Adrián Regos that considerably improved the manuscript.

Additional Information and Declarations

Competing Interests

Author Contributions

Animal Ethics

Data Availability

ICNF, SPEA and REN Biodiversity Chair were involved in the study design and data collection. Collaborators of the Liga para a Proteccão da Natureza, Quercus–Associacão Nacional de Conservacão da Natureza, Sociedade Portuguesa para o Estudo das Aves, LABOR–Laboratório de Ornitologia/University of Évora participated during the surveys. The authors declare there are no competing interests.

João P. Silva and Francisco Moreira conceived and designed the experiments, performed the experiments, analyzed the data, wrote the paper, prepared figures and/or tables, reviewed drafts of the paper.

Ricardo Correia performed the experiments, analyzed the data, prepared figures and/or tables, reviewed drafts of the paper.

Hany Alonso, Ricardo C. Martins, Marcello D’Amico, Ana Delgado, Hugo Sampaio and Carlos Godinho performed the experiments, reviewed drafts of the paper.

The following information was supplied relating to ethical approvals (i.e., approving body and any reference numbers):

This work dealt with surveys and was carried out in partnership with the Portuguese national authority for Nature Conservation (ICNF—Instituto da Conservação da Natureza e das Flotrestas). It implied no animal manipulation or experimentation.

The following information was supplied regarding data availability:

The raw data concerning the results (number of males counted, density and habitat availability) obtained in each survey area is provided as a Data S1.

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
