# Peer review of "EU protected area network did not prevent a country wide population decline in a threatened grassland bird"

_PeerJ, doi:10.7717/peerj.4284_

## Round 0.1 · original submission · Major Revisions

As you will see, both reviewers have a very different opinion. I think the study has important implications for the conservation of the Little Bustard,and can help to its conservation. Therefore you should read carefully the 2nd Reviewer's opinion and try to rebut the considerations they have made, especially the questions on your methods. Also Rev 1. has made also very useful suggestions that will improve the paper. Despite the different opinions I consider your paper can be a good contribution if you address their suggestions.

·

Basic reporting

no comment

Experimental design

no comment

Validity of the findings

no comment

Additional comments

Dear authors,

I really enjoyed reading the manuscript. The study has important implications for the conservation of the Little Bustard, and it can help support the reporting obligations of Portugal in relation to the Bird Directive. It also shows how management policies should be targeted at maintaining not only habitat availability but also habitat quality within protected areas, and highlights the need of going further from a mere designation of protected areas if an effective conservation wants to be achieved. I consider the data robust and the statistical analysis is sound.

Therefore, I recommend accepting the manuscript after addressing a few minor issues:

Minor changes

Line 31. “within the species remaining European stronghold”. Sorry, but it is not clear to me what the authors mean. The authors could indicate that Little Bustard is included in the Annex I of the Bird Directive, which is clearly related to the objectives of SPAs.

Lines 32. Please remove “(the European Union network of protected areas)”. Natura 2000 has been already defined in line 30.

Line 34-35. These SPAs are designated only for farmland bird conservation? I would say that SPAs are designated under the EU Bird Directive to protect bird species listed in the Annex I. In general SPAs are aimed at protecting also other species that do not have to be necessarily farmland species. Also, I am wondering if these “30 areas without EU protection” can be under national or regional protection. Please clarify it.

Line 36. Awkward sentence. Please rephrase, e.g. “the bustard national population declined 49% over the last 10-14 years, especially outside SPA.” Please indicate the population trends inside and outside the SPA system, or the difference between within and outside to show more explicitly the degree of effectiveness of SPAs.

Keywords: I would suggest “steppe birds” instead of ‘farmland bird’ which is already mentioned in the abstract

Introduction

Lines 57. Can be the author a bit more specific? Some examples of these biodiversity values? Do the authors refer to e.g. habitat or climate suitability?

Lines 58. I mostly agree, but instead of using Cabeza (2013) as reference (there are many others), I would show one of the few studies that uses population trends as follows: “(but see Pellissier et al., 2013).”

Reference:
Pellissier, V., Touroult, J., Julliard, R., Siblet, J. P., & Jiguet, F. (2013). Assessing the Natura 2000 network with a common breeding birds survey. Animal Conservation, 16(5), 566-574.

Line 85. This reference is very old, I would suggest adding a more recent reference to show the last trends for this species. According to BirdLife International (2016) “The European population is estimated to be declining by 30-49% in three generations (30.9 years)”.

Reference:
BirdLife International. 2016. Tetrax tetrax. The IUCN Red List of Threatened Species 2016: e.T22691896A90095419. http://dx.doi.org/10.2305/IUCN.UK.2016-3.RLTS.T22691896A90095419.en. Downloaded on 07 November 2017.

Line 170. What do the authors mean by “proportionally”?

Appendix S1 seems to be incomplete (see last sentence). “Because these IBAs were not subjected to any sort of management, for the …” Please revise it.

Discussion

Although the discussion and the main conclusions are well stated and linked to the research questions, previous studies have already showed that field management and timing can play a key role in the aptitude of fallows for steppe birds such as Little Bustard during the breeding season, affecting the vegetation structure as well as the amount and type of food resources at microhabitat level. For instance, early herbicide application and shredding were found to be the best treatments for Little Bustard in steppe areas of NE Spain (Robleño et al 2017). Thus, to improve the effectiveness of management actions, conservation guidelines for farmland birds should consider microhabitat preferences, given the importance not only of the amount of habitat provided but also of vegetation structure and food availability. I think that these issues deserve attention and could be mentioned in the discussion to better understand the relevance of habitat quality (and not only availability) for the conservation of Little Bustard in particular, and farmland species in general. In addition, previous research also highlighted the importance of intraspecific interactions in the definition of the habitat selection pattern of females and families (Tarjuelo et al 2013), issues also relevant for the conservation of the species.

Ponjoan et al (2012) can help to support the statement in lines 238-239, these authors found that ranging behaviour of males could be partially explained by age, habitat quality and site.

References

Ponjoan, A., Bota, G., & Mañosa, S. (2012). Ranging behaviour of little bustard males, Tetrax tetrax, in the lekking grounds. Behavioural processes, 91(1), 35-40.

Robleño, I., Bota, G., Giralt, D., & Recasens, J. (2017). Fallow management for steppe bird conservation: the impact of cultural practices on vegetation structure and food resources. Biodiversity and Conservation, 26(1), 133-150.

Tarjuelo, R., Delgado, M. P., Bota, G., Morales, M. B., Traba, J., Ponjoan, A., ... & Mañosa, S. (2013). Not only habitat but also sex: factors affecting spatial distribution of Little Bustard Tetrax tetrax families. Acta ornithologica, 48(1), 119-128.

Reviewer 2 ·

Basic reporting

The manuscript seeks to evaluate the efficiency of Natura 2000 areas in truly protecting habitat and encompassed biodiversity. The authors show a remarkable decrease in the population size of the little bustard (Tetrax tetrax) in Southern Portugal, one of the most important areas for the conservation of the species. Tough results are valuable and discouraging, I consider that they only constitute an initial step in the more profound and important evaluation that the authors propose. Fundamentally, I found as main problems that: (1) collected data and used methods do not allow to properly evaluate a trend of the population size of this or any other species (only two periods, 2003-2006 and 2016 were considered), (2) causal factors of changes in the phenomenon under study are not evaluated and poorly discussed (e.g. climatic conditions, hunting pressure, farmland use), except for the area under grasslands. (3) the area under grasslands is analyzed as a dependent variable, and not as a causal factor.

I acknowledge that data collection can be too expensive in economic and human effort terms, but alternatives exist to achieve to a more exhaustive analysis (e.g. by reducing the number of areas sampled but increasing temporal sampling, by exploring the land use or cover trends). I hope that my rejection decision does not discourage the authors since I found that they have many options to improve their findings and finally manage in a more efficient way these protected areas.

Experimental design

no comment

Validity of the findings

no comment

Additional comments

no comment

---

## Round 0.2 · accepted · Accept

The authors have justified criticism and included all comments and changes suggested by the reviewers and the editor.